# Nebulized CLODOS Technology Shows Clear Virucidal Properties against the Human Coronavirus HCoV-229E at Non-Cytotoxic Doses

**DOI:** 10.3390/v13030531

**Published:** 2021-03-23

**Authors:** Sabina Andreu, Inés Ripa, Raquel Bello-Morales, José Antonio López-Guerrero

**Affiliations:** 1Departamento de Biología Molecular, Universidad Autónoma de Madrid, Cantoblanco, 28049 Madrid, Spain; ines.ripa@cbm.csic.es (I.R.); raquel.bello-morales@uam.es (R.B.-M.); ja.lopez@uam.es (J.A.L.-G.); 2Centro de Biología Molecular Severo Ochoa, Spanish National Research Council—Universidad Autónoma de Madrid (CSIC-UAM), Cantoblanco, 28049 Madrid, Spain

**Keywords:** virucidal, nebulization, HCoV-229E, SARS-CoV-2

## Abstract

The emergent human coronavirus SARS-CoV-2 and its high infectivity rate has highlighted the strong need for new disinfection systems. Evidence has proven that airborne transmission is an important route of spreading for this virus. Therefore, this short communication introduces CLODOS Technology^®^, a novel strategy to disinfect contaminated surfaces. It is a product based on stable and 99% pure chlorine dioxide, already certified as a bactericide, fungicide and virucide against different pathogens. In this study, CLODOS Technology^®^, by direct contact or thermonebulization, showed virucidal activity against the human coronavirus HCoV-229E at non-cytotoxic doses. Different conditions such as nebulization, exposure time and product concentration have been tested to standardize and optimize this new feasible method for disinfection.

## 1. Introduction

The Coronaviridae are a large family of lipid-enveloped, single-stranded positive-sense RNA viruses that can cause respiratory, gastrointestinal, hepatic and neurological problems both in humans and animals [1]. Several human coronaviruses (HCoVs) are responsible for generally mild upper respiratory tract infections and the common cold, such as HCoV-NL63, HCoV-229E, HCoV-OC43 and HCoV-HKU1 [2]. Two of the more virulent coronaviruses, severe acute respiratory syndrome coronavirus (SARS-CoV) and Middle East respiratory syndrome coronavirus (MERS-CoV), emerged in human populations in 2003 and 2012, respectively [3]. Recently, the emergence and rapid global spread of the new coronavirus SARS-CoV-2, identified as the causal agent of coronavirus disease 2019 (COVID-19), led the WHO to declare a state of pandemic in March 2020, with more than 100 million confirmed cases and over 2 million deaths, at the time of writing this report [4].

Despite the extraordinary speed at which several vaccines against SARS-CoV-2 have been developed and approved [5], the lack of effective pharmacological treatments, the adaptability of coronaviruses to other hosts, including humans, and therefore the potential of these viruses to cause epidemics, [6], makes the search for new systems to limit the infection strongly necessary].

According to the WHO, SARS-CoV-2 can be transmitted via aerosols, larger respiratory droplets and fomites (surface deposits) [7]. Airborne transmission or aerosols are produced when particles named “droplet nuclei” (smaller than 5–10 µm in diameter) remain suspended in the air for a variable period of time [8] and may be transported over considerable distances [9]. These aerosols can be generated from normal breathing and also from the evaporation of larger droplets produced by coughing and sneezing by both asymptomatic and symptomatic people [10,11]. Consequently, there is strong evidence that airborne transmission is a significant factor in the spread of SARS-CoV-2, and that the accumulation of infectious bioaerosols in spaces with poor ventilation contributes to the transmission of the virus [11,12]. Although face masks and social distancing help to reduce this route of transmission, further precautions specific to aerosols are required.

This report presents CLODOS Technology^®^, a novel method to inactivate human coronaviruses such as HCoV-229E (α-CoV) in aerosol form and on surfaces, based on pure and stable chlorine dioxide. CLODOS Technology^®^ is an aqueous solution where chlorine dioxide gas has been generated at a certified concentration, with a purity of 99.9%, and stability over time, ensuring the absence of by-products during the disinfection process [13,14]. CLODOS Technology^®^ is already certified as a bactericide, fungicide and virucide against different pathogens (see Appendix A, Figure A1) and can be applied in any type of environment and for surface disinfection, even in the presence of people, based on the certifications obtained. CLODOS Technology^®^ works not only through direct contact but can also be nebulized in closed spaces by using a special thermonebulizer machine. This method, which is currently under study, allows the disinfection of the air by applying the product through misting. It has been discovered that the antimicrobial activity of chlorine dioxide is due to oxidative modification of the tryptophan and tyrosine residues of certain proteins, leading to their denaturation [15,16]. Regarding SARS-CoV-2, since its essential spike protein (S) contains 54 tyrosine and 12 tryptophan residues, it is assumed that these residues would react with chlorine dioxide and produce a rapid viral inactivation [17].

Chlorine dioxide has been in use for decades in several fields, such as the treatment of public water supplies, the food and paper industries, and even during health emergencies [18]. For instance, chlorine dioxide was used to decontaminate anthrax spores from the 2001 US terrorist attacks [19]. However, the obtention of pure and stable chlorine dioxide (CLODOS Technology^®^) was achieved less than 20 years ago, and it is now considered as a revolutionary product in the industrial market due to its multitude of new and advantageous applications. Nonetheless, practically most of the state of the art technology that is available is not adequate with current chlorine dioxide technology, that assures purity and stability; traditional chlorine dioxide techniques do not achieve the adequate purity and have multiple by-products in their dilution [14].

Different methods for surface disinfection against SARS-CoV-2 are currently under study, such as ultraviolet irradiation, chemical disinfectants and ozone (Cristiano, 2020). The novel feature that CLODOS Technology^®^ presents compared to these methods is its safety term 0, which means that the disinfected space can be immediately occupied by people [14].

In the present study, the virucidal effect of CLODOS Technology^®^ against HCoV-229E was tested in different conditions. Firstly, a cytotoxicity assay was performed to affirm that the product does not have any toxic effect in its nebulized form at the doses suggested for use by the manufacturer. Virucidal assays were performed with directly sprayed or nebulized product, showing that CLODOS Technology^®^ is a potential strategy for the disinfection of contaminated surfaces or aerosols.

## 2. Materials and Methods

### 2.1. Cell Cultures

The Huh-7 cell line [20] was generously provided by Dr. Sonia Zúñiga, from the Spanish National Centre for Biotechnology. This cell line was cultured in culture medium (CM) containing low-glucose Dulbecco’s modified Eagle medium (DMEM) (Life Technologies, Paisley, UK) supplemented with 10% fetal bovine serum (FBS), penicillin (50 U/mL) and streptomycin (50 μg/mL) at 37 °C in a humidified atmosphere containing 5% CO_2_.

### 2.2. Viruses

HCoV-229E expressing a GFP reporter protein was generously provided by Dr. Volker Thiel, from the University of Bern. This virus was propagated on Huh-7 cells for 5 days at 33 °C with 5% CO_2_. The infectious titer of the virus stocks was determined according to the Reed and Muench formula on Huh-7 cell monolayers [21].

### 2.3. Reagents

The product CLODOS Technology^®^ is defined as a yellow aqueous solution based on pure and stable chlorine dioxide (CAS Nr. 10049-04-4) and was provided by STC Company (Servicios Técnicos de Canarias S.L, Las Palmas de Gran Canaria, Spain). The thermonebulization device Clodos Jetechnology^®^, which provides a dry fogging, achieving the high and rapid volatility of the gas dissolved in the disinfectant but avoiding any technical incompatibility (see Appendix B, Figure A2), was provided and developed by STC Company and INTIAI ECO (Madrid, Spain). The hermetic nebulization container (volume of 15 L) where assays were performed was also provided by INTIAI ECO. Low-glucose DMEM and FBS were purchased from Sigma Chemical Co. (St. Louis, MO, USA).

### 2.4. Quality and Concentration Control of CLODOS Technology^®^

For the traceability of the results, the quality and concentration control of CLODOS Technology^®^ product is mandatory in each test performed. For this, the KEMIO^™^ measurement platform with the serial number 10D07190003 (Palintest, Halma, UK) was used. This device works with a measurement system based on chronoamperometry, an electrochemical method approved by the United States Environmental Protection Agency (EPA), which measures current flow when a fixed voltage is applied to the calibrated electrode (chlorine dioxide-calibrated electrodes KEM25CDX were used).

With this method, high accuracy can be obtained because interference from chlorite and other chemical substances is avoided, which is not possible with other traditional colorimetric methods such as DPD (diethyl-p-phenylene diamine).

### 2.5. Cell Viability Assay

The cytotoxic effects of nebulized CLODOS Technology^®^ in Huh-7 cells were analyzed by the MTT method using the kit Cell Titer 96^®^ Non-Radioactive Cell Proliferation Assay from Promega (Madison, WI, USA). Previous certified assays have assured bactericidal and virucidal activity of direct-sprayed CLODOS Technology^®^ at a concentration of 100 ppm without any cytotoxic effect, so the purpose of this study was to test the cytotoxicity of the product in its nebulized form at higher doses. Non-confluent monolayers of cells plated in 96-well tissue culture dishes were placed in the nebulization container and nebulized for 15 s with CLODOS Technology^®^ at 225 and 450 ppm. Cells were exposed to the nebulization cloud inside the hermetic container for 1 and 5 min. Then, cells were incubated for 24 h at 37 °C in a humidified atmosphere containing 5% CO_2_. Subsequently, cells were incubated with a final concentration of 0.5 mg/mL of MTT in a humidified atmosphere for 4 h, at which point formazan crystals were solubilized in 10% SDS in 0.01 M of HCl. The resulting-colored solution was quantified using a scanning multiwell spectrophotometer (ELISA reader), measuring the absorbance of formazan at 595 nm.

### 2.6. Virucidal Effect of CLODOS Technology^®^

HCoV-229E at a known titration was plated in 24-well tissue culture dishes. The amount of viral solution deposited in each well was 80 microlitres, covering the entire surface of the plate well. Then, the plate was placed in the nebulization container and nebulized or mock-nebulized with CLODOS Technology^®^, using the thermonebulization device Clodos Jetechnology^®^, at room temperature. Different conditions were tested: (i) direct contact spray between the product and the virus, by adding 500 µL of the product diluted in CM at a concentration of 150 ppm to each well, (ii) 1 s nebulization and 1, 5 and 15 min of exposure to the product at 225 ppm and (iii) 1, and 2 s nebulization with 5 min of exposure to the product at 450 ppm. Right after the treatment with the product, 30 µL were collected from each sample and resulting viral titer was calculated in Huh-7 cells according to the endpoint dilution assay (see Section 2.7).

### 2.7. Endpoint Dilution Assay

Non-confluent monolayers of Huh-7 cells were plated in 96-well tissue culture dishes and cultured in CM. Serial dilutions (10-1 to 10-7) of HCoV-229E were prepared and inoculated onto replicate cell cultures. Cells were then incubated at 33 °C in a humidified atmosphere containing 5% CO_2_ for 72 h. Finally, the 50% tissue culture infectious dose per mL (TCID50/mL) was determined, considering the final dilution that showed cytopathic effect and calculated using the Reed and Muench method [21].

### 2.8. Statistics

Three biological replicates were performed for each assay. A Student’s *t*-test was performed for independent measures to compare the mean values of each data set, with *p*-values < 0.05 being categorized as significant.

## 3. Results

### 3.1. CLODOS Technology^®^ Is Non-Toxic in Huh-7 Cells at 225 ppm

To study the cytotoxicity of nebulized CLODOS Technology^®^, Huh-7 cells were cultured in CM and nebulized for 15 s with the product at a concentration of 225 or 450 ppm. Cells were left in the presence of the nebulization atmosphere for 1 and 5 min. A group of cultured wells were non-treated and left as control. The cytotoxicity assay revealed that cell viability remains high in treated cells (over 95%) (Figure 1). Huh-7 viability did not decrease significantly when the exposure time varied between 1 and 5 min.

### 3.2. Virucidal Effect of CLODOS Technology^®^

#### 3.2.1. Directly Sprayed CLODOS Technology^®^ Shows Virucidal Activity against HCoV-229E

The first condition tested was the virucidal effect of CLODOS Technology^®^ in direct contact with HCoV-229E. The virus was plated and directly sprayed with 500 µL of the product at a concentration of 150 ppm. Compared to the untreated control, the viral titer decreased by 95.5% when HCoV-229E was subjected to direct spray with CLODOS Technology^®^, a decrease of more than one order of magnitude, statistically significant (Figure 2).

#### 3.2.2. Virucidal Effect of Nebulized CLODOS Technology^®^ at Different Conditions

The potential virucidal effect of nebulized CLODOS Technology^®^ against HCoV-229E was also tested. The virus was plated and placed in the nebulization container and this space was nebulized for 1 s with CLODOS Technology^®^ at 225 ppm. Then, the samples were exposed to this nebulization cloud for 1, 5 or 15 min. After 1 s of nebulization and 1 min of exposure, the viral titer decreased by 42.35% in comparison to the non-nebulized control. When the exposure time was extended to 5 min, the viral titer dropped even more, up to a total decrease of 78.45%. From then on, viral titer seemed not to change (Figure 3).

Once the optimal exposure time was set to 5 min, one more assay was performed to study the effect of increasing the nebulization time to 2 s while maintaining the exposure time. In this experiment, the concentration of CLODOS Technology^®^ was raised to 450 ppm, reaching a significant reduction in the viral titer of at least 80%. When nebulization lasted 1 s, viral titer decreased 76.65%. By increasing the nebulization time to 2 s, viral titer decreased by more than 90%, which corresponds to one order of magnitude (Figure 4).

Finally, optimal conditions for the disinfection of such titer of HCoV-229E in the nebulized container used were set to CLODOS Technology^®^ nebulized at a concentration of 450 ppm, with a nebulization time of 2 s and exposure time of 5 min.

## 4. Discussion

Preventive measures such as hand washing, face mask use, ventilation and social distancing are useful to limit the spread of SARS-CoV-2, but they are not sufficient. Poorly ventilated and closed spaces, coupled with the emission of bioaerosols when talking or coughing, increase the probability of transmission of airborne pathogens [22]. CLODOS Technology^®^ is a wide-spectrum disinfectant that, unlike other disinfectant methods based on ozone or UV-light [23], does not require security time, so potentially, and based on certifications obtained, it may be applied on surfaces and spaces in the presence of people.

The MTT assay demonstrated the absence of cytotoxicity of the nebulized CLODOS Technology^®^ at 225 and 450 ppm. Moreover, when the product was directly sprayed at 150 ppm, it achieved a reduction in the viral titer of 95.5% (Figure 2). Therefore, the product could be a great alternative in the direct cleaning of surfaces of any material.

In terms of nebulization, CLODOS Technology^®^ nebulized at 225 ppm reduced the viral titer almost by 80% with 1 s of nebulization and 5 min of exposure time. After 5 min of exposure, the viral titer did not decrease proportionally to what was observed in the previous conditions (Figure 3). This outcome might be explained by the fact that all the nebulized product came into contact and inactivated the viral particles in a 1:1 ratio, but there were still viral particles in the sample. For this reason, in the subsequent experiment, the nebulization time was raised, and the exposure time was set to 5 min, maintaining the same viral charge. Moreover, when the product was used at a double concentration (450 ppm), the viral titer significantly decreased by one order of magnitude.

The optimal conditions for the nebulized container in which experiments were made were set to CLODOS Technology^®^ nebulized at a concentration of 450 ppm, a nebulization time of 2 s and exposure time of 5 min, when 91.8% of viral reduction was obtained. These conditions should be reviewed and extrapolated to larger volumes, such as school classrooms, hospital rooms or other closed spaces. Furthermore, it would be also interesting to unravel the mechanism of the inactivating action of the product and its persistence in different materials and in the air, as little is still known about these aspects.

It must be taken into account that these tests have been carried out starting from a stock of HCoV-229E with a high titer, and that, on average, the viral load found in aerosols or on surfaces may be significantly similar [24]. Prior experiments showed that 1 min of loud speaking could produce thousands of droplet nuclei per second, each containing approximately 1000 viral particles, starting from an average viral load of 7 × 10^6^ per mL [25,26]. However, recent studies reveal that there is a 0.37% probability that a 10 µm droplet contains at least one infective unit, prior to dehydration [27], and that very few particles actually carry infective viruses [9].

HCoV-229E and SARS-CoV-2 belong to different subfamilies (α-CoV and β-CoV, respectively) and use different receptors to enter the target cell [28]. Nonetheless, as chlorine dioxide seems to react with the essential spike protein (S) and the lipid envelope [17], this technology may be useful to inactivate other coronaviruses as well. 

In conclusion, our results indicate that disinfection with wide-spectrum thermonebulized CLODOS Technology^®^ was effective at reducing the viral titer of HCoV-229E. The recommended applications for this technology are: (i) the direct cleaning of surfaces with product at a concentration of 150 ppm and (ii) nebulization in closed spaces with a Clodos Jetechnology^®^ nebulizer device at 225 ppm. Further research needs to be performed to apply this new technology in the disinfection of contaminated surfaces and bioaerosols containing other coronaviruses, such as SARS-CoV-2.

## Figures and Tables

**Figure 1 viruses-13-00531-f001:**
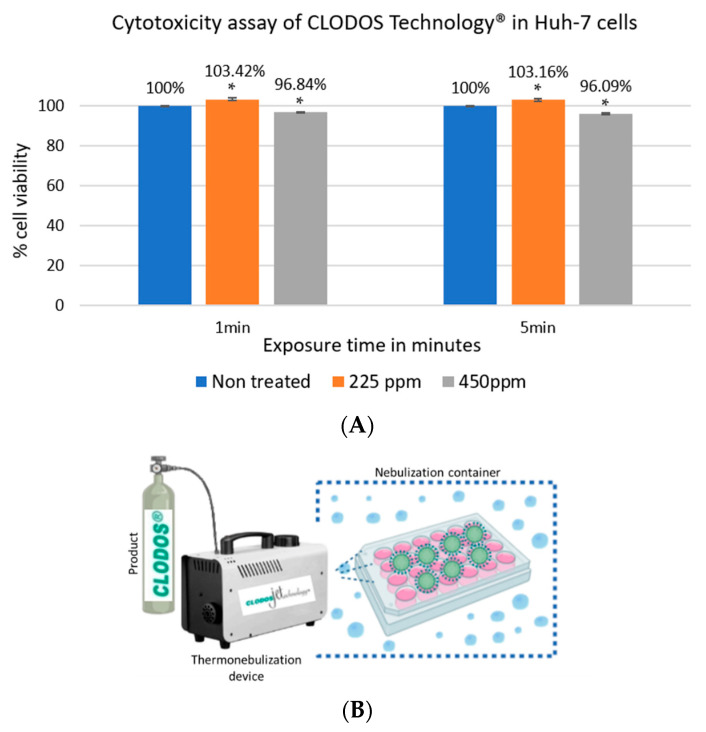
(**A**) Viability of Huh-7 cells nebulized with CLODOS Technology^®^. Huh-7 cells cultured in culture medium (CM) were nebulized or mock-nebulized with CLODOS Technology^®^ at 225 ppm and 450 ppm for 15 s. Different overall exposure times were tested: 1- and 5-min. Cell viability was then measured by MTT tetrazolium salt assay and calculated as the percentage of viability compared to the untreated cells; columns represent the mean viability ± S.D (*n* = 4), after exposure to the product; * *p* < 0.05. (**B**) Nebulization protocol diagram. HCoV-229E was placed in 24-well culture plates. The plates were then inserted into the nebulizer container. The thermonebulization apparatus CLODOS Jetechnology^®^ was filled with the product and the plate was nebulized for the stated times.

**Figure 2 viruses-13-00531-f002:**
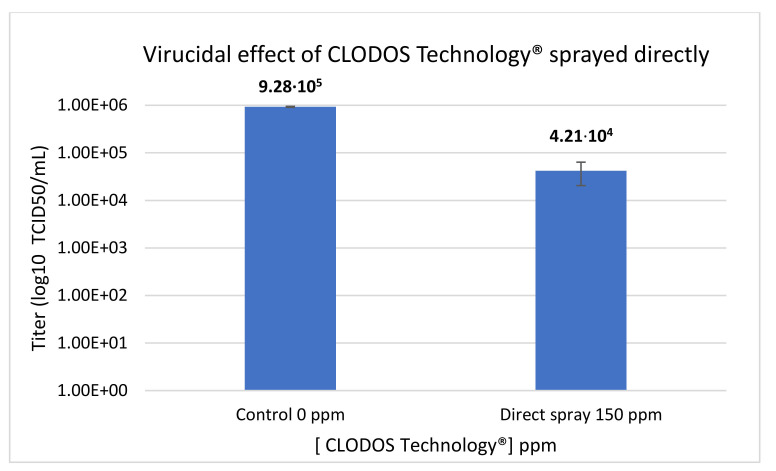
Virucidal effect of CLODOS Technology^®^ sprayed directly against HCoV-229E. HCoV-229E was subjected to 150 ppm of directly sprayed CLODOS Technology^®^. Viral titer (50% tissue culture infectious dose per mL (TCID50/mL)) was determined immediately after the contact by the endpoint dilution assay. Columns represent the mean TCID50/mL ± S.D (*n* = 3), after exposure to the product.

**Figure 3 viruses-13-00531-f003:**
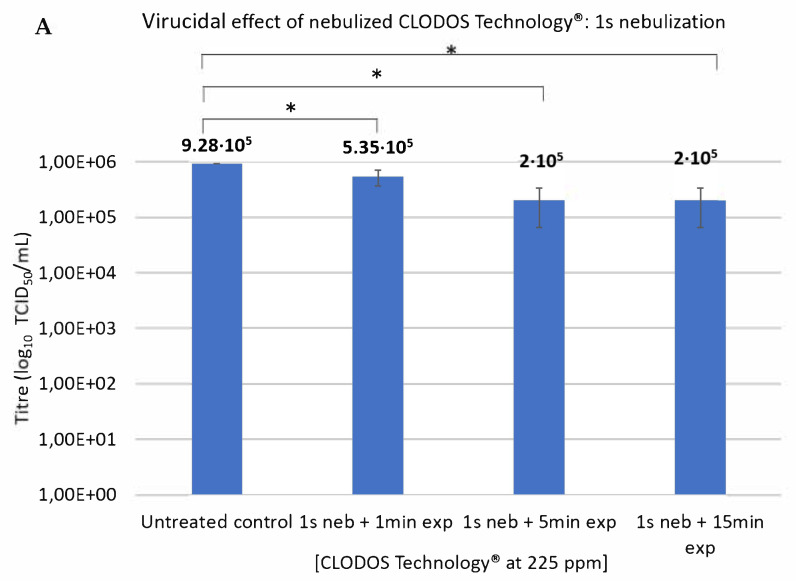
Virucidal effect of nebulized CLODOS Technology^®^ against HCoV-229E after 1 s of nebulization and 1, 5 or 15 min of exposure. HCoV-229E was subjected to nebulization with CLODOS Technology^®^ at a concentration of 225 ppm. The nebulization time was set to 1 s and the exposure times to 1, 5 or 15 min. Viral titer (TCID50/mL) was determined immediately after completion of exposure time by the endpoint dilution assay. (**A**) Bar graph where columns represent the mean TCID50/mL ± S.D (*n* = 3), after exposure to the product. Neb: nebulization, Exp: exposure * *p* < 0.05. (**B**) Table that shows the viral titer and the percentage of decrease in viral titer compared to the untreated control, for each condition.

**Figure 4 viruses-13-00531-f004:**
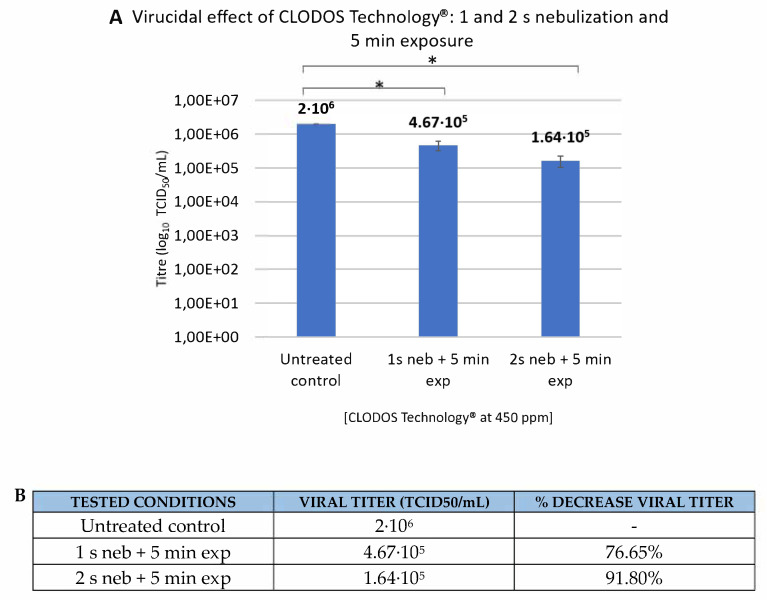
Virucidal effect of nebulized CLODOS Technology^®^ against HCoV-229E after 1 or 2s of nebulization and 5 min of exposure. HCoV-229E was subjected to nebulization with CLODOS Technology^®^ at a concentration of 450 ppm. The nebulization times that were tested were 1 and 2 s, and the exposure time was set to 5 min. Viral titer (TCID50/mL) was determined immediately after the spray by the endpoint dilution assay. (**A**). Bar graph where columns represent the mean TCID50/mL ± S.D (*n* = 3), after exposure to the product. Neb: nebulization, Exp: exposure * *p* > 0.05; (**B**). Table that shows the viral titer and the percentage of decrease in viral titer compared to the untreated control, for each condition.

## Data Availability

No new data were created or analyzed in this study. Data sharing is not applicable to this article.

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
