# Peer review of "Nebulized CLODOS Technology Shows Clear Virucidal Properties against the Human Coronavirus HCoV-229E at Non-Cytotoxic Doses"

_viruses, 2021, doi:10.3390/v13030531_

Round 1

Reviewer 1 Report

Dear authors, 

I read with interest the paper of Andreu et al. concerning the new CLODOD technology as a technique to inactivate the HCoV-229E coronavirus. My remarks are as follows:

In the result section: would it be possible to add the p value to the statement that Huh-8 variability was not significantly decreased.

p value often stated that it was less than 0.05; however it would be preferable to report the exact p value.

In the figures (eg. 3B, 4A, 4B), add the p value compared to the control condition. Currently it is not clear what is compared to what.

After reading the manuscript, one question remains: is a cumulative dose safe as well? How often should this be repeated in a certain room?

Author Response

Dear reviewer,

Thanks in advance for reading our manuscript. Regarding your corrections:

-Point 1. p values in viability assay. We added p values to the MTT figure (viability assay). Figure 1

-Point 2. Exact p-value. The T-test calculator that we use to calculate t and p values gives us the exact p-values for each means. Nonetheless, in short communications, figures tend to present p values in this style (e.g. p <0.01 or p<0.05). If it is possible, we would prefer to maintain this format.

-Point 3. p-value compared to the control. We changed figures 3A and 4A and added the p-value compared to the control condition, so now it can be better understood.

-Point 4. Cumulative dose. This is an interesting question. The CLODOS product, like bioaerosols, falls to surfaces, degrades over time, and leaves no residue. The persistence of the product depends on different characteristics, such as the presence of organic matter, light, temperature and ventilation. The product is photosensitive and above 10 degrees of temperature, its degradation speed increases. 

According to the manufacturer, CLODOS sprayed directly shows no harmful effects when sprayed consecutively. With regard to the nebulized product, it is degraded at a certain time and new product can be expelled. 

Our experiments were carried out in a 15 liter volume airtight container. These results can be extrapolated to larger rooms (with known volume), where the dosage of the product will not be continuous. The purpuse of the product consists of immediate disinfection, without the need to be continuously nebulizing.

Please, if you have any questions, do not hesitate to contact me again.

Reviewer 2 Report

The manuscript presented by Andreu and colleagues  is interesting and covers a hot topic due to the global emergency caused by SARS-CoV-2.

However, I believe that the authors should give some clarification about the work.

-Authors use HCoV-229E. I suggest to better introduce this coronavirus and its effects in human and the existent therapeutic approaches

-Authors claim that this potential therapeutic strategy could be useful to fight SARS-Cov-2. Accordingly to affirm this authors should compare the structural components of two viruses and the mechanism of action and infection of both. To this end, it is not clear what is the target of the therapeutics for inactivating  HCoV-229E. After this clarification the analysis about the possible application against SARS-CoV-2 can be performed.

-Authors should discuss possible limitation of the method

-Finally although the data is interesting an in vivo study on animals infected with HCoV-229E will give a complete preclinical evaluation of this potential therapeutic agent.

Author Response

Dear reviewer,

Thank you in advance for reviewing our manuscript. About your corrections:

-Point 1. Introducing coronavirus HCoV-229E: We present in lines 26-28 HCoV-229E as a type of human coronavirus that causes generally mild upper respiratory tract infections and common cold. We decided not to talk extensively about its symptoms and treatments because the product that we present has not ANTIVIRAL action; we test its activity as VIRUCIDAL. Therefore, CLODOS Technology is not meant to be used as treatment/therapy against HCoVs, as it is recommended as disinfectant of surfaces and air.

-Point 2. Comparison between SARS-CoV-2 and HCoV-229E. We added lines 261-264 to clearify this: 

Line 261: "HCoV-229E and SARS-CoV-2 belong to different subfamilies (α-CoV and β-CoV, respectively) and use different receptors to enter the target cell [28]. Nonetheless, as chlorine dioxide seems to react with the essential spike protein (S) and the lipid envelope [17], this technology may be useful to inactivate other coronaviruses as well."

Point 3- Limitations of the method. We added the next paragraphs:

Line: 250: "These conditions should be reviewed and extrapolated to larger volumes, such as school classrooms, hospital rooms or other closed spaces. Furthermore, it would be also interesting to unravel the mechanism of action of the product and its persistence in different materials and in the air, as little is still known about these aspects."

Line 271: "Further research needs to be performed to apply this new technology in the disinfection of contaminated surfaces and bioaerosols containing other coronaviruses, such as SARS-CoV-2."

Point 4- In vivo study of animals infected. The product is tested as virucidal, not as antiviral, so it is not meant to be therapeutic. Consequently, an in vivo study of animals infected with the virus is not proposed, as the product will be used as disinfectant.

Please, if you have any questions, do not hesitate to contact me again.

Round 2

Reviewer 2 Report

Authors addressed my concerns, unfortunately at a first reading I supposed this agent as antiviral. I apologize for this, accordingly, current virucidal agents should be compared with this product. After this comparison the manuscript can be published

Author Response

Dear reviewer,

Thank you for letting us know. We have added the next paragraph:

Line 77: "Different methods for surface disinfection against SARS-CoV-2 are currently under study, such as ultraviolet irradiation, chemical disinfectants and ozone (Cristiano, 2020). The novel feature that CLODOS Technology® presents compared to these methods is its safety term 0, which means that the disinfected space can be immediately occupied by people [14]."